# Biocompatibility of ABS and PLA Polymers with Dental Pulp Stem Cells Enhance Their Potential Biomedical Applications

**DOI:** 10.3390/polym15244629

**Published:** 2023-12-06

**Authors:** Fabiane Barchiki, Letícia Fracaro, Alejandro Correa Dominguez, Alexandra Cristina Senegaglia, Isadora May Vaz, Paulo Soares, Sérgio Adriane Bezerra de Moura, Paulo Roberto Slud Brofman

**Affiliations:** 1Core for Cell Technology, School of Medicine, Pontifícia Universidade Católica do Paraná (PUCPR), Curitiba 80215-901, Brazil; leticiafracaro@gmail.com (L.F.); acsenegaglia@hotmail.com (A.C.S.); isadora.may@pucpr.br (I.M.V.); paulo.brofman@pucpr.br (P.R.S.B.); 2INCT—REGENERA National Institute of Science and Technology in Regenerative Medicine, Rio de Janeiro 21941-902, Brazil; 3Laboratory of Basic Biology of Stem Cells, Carlos Chagas Institute, Fiocruz-PR, Curitiba 81350-010, Brazil; alejandro.correa@fiocruz.br; 4LaBES—Laboratory of Biomaterials and Surface Engineering, Polytechnic School, Pontifícia Universidade Católica do Paraná (PUCPR), Curitiba 80215-901, Brazil; pa.soares@pucpr.br; 5Departament of Morphology, Campus Universitário Lagoa Nova, Universidade Federal do Rio Grande do Norte (UFRN), Natal 59072-970, Brazil; sergioabm@gmail.com

**Keywords:** PLA, ABS, DPSCs, 3D printing, osteogenic differentiation, genetic stability, stem cell, regenerative medicine

## Abstract

Polylactic Acid (PLA) and Acrylonitrile–Butadiene–Styrene (ABS) are commonly used polymers in 3D printing for biomedical applications. Dental Pulp Stem Cells (DPSCs) are an accessible and proliferative source of stem cells with significant differentiation potential. Limited knowledge exists regarding the biocompatibility and genetic safety of ABS and PLA when in contact with DPSCs. This study aimed to investigate the impact of PLA and ABS on the adhesion, proliferation, osteogenic differentiation, genetic stability, proteomics, and immunophenotypic profile of DPSCs. A total of three groups, 1- DPSC-control, 2- DPSC+ABS, and 3- DPSC+PLA, were used in in vitro experiments to evaluate cell morphology, proliferation, differentiation capabilities, genetic stability, proteomics (secretome), and immunophenotypic profiles regarding the interaction between DPSCs and polymers. Both ABS and PLA supported the adhesion and proliferation of DPSCs without exhibiting significant cytotoxic effects and maintaining the capacity for osteogenic differentiation. Genetic stability, proteomics, and immunophenotypic profiles were unaltered in DPSCs post-contact with these polymers, highlighting their biosafety. Our findings suggest that ABS and PLA are biocompatible with DPSCs and demonstrate potential in dental or orthopedic applications; the choice of the polymer will depend on the properties required in treatment. These promising results stimulate further studies to explore the potential therapeutic applications in vivo using prototyped polymers in personalized medicine.

## 1. Introduction

Mesenchymal Stem Cells (MSCs) are adult stem cells capable of differentiating into diverse cell types, including osteocytes, adipocytes, and chondrocytes [1,2]. Although MSCs were first found in bone marrow, these stem cells have since been isolated from various other sources, such as adipose tissue, umbilical cord blood, placenta, and, more recently, dental tissues [3]. Dental Pulp Stem Cells (DPSCs) were first described by Gronthos et al. in 2000 [4], and they are derived from dental pulp, the soft tissue inside the tooth. These cells demonstrate a strong potential for odontogenic differentiation, making them excellent candidates for dental tissue engineering, including dentin and pulp tissue regeneration [5,6]. Along with their odontogenic capabilities, DPSCs have shown osteogenic potential and can differentiate into osteoblast-like cells, contributing to bone formation [7,8].

One of the critical advantages of DPSCs is their accessibility; they can be obtained through non-invasive procedures from extracted teeth, such as wisdom teeth, which is advantageous in clinical settings. Coupled with their high proliferation rates, which means even a small sample can potentially yield many cells for therapeutic use, these characteristics suggest that DPSCs hold immense promise for regenerative medicine, particularly in applications concerning dental and bone tissue regeneration. The versatile properties and potential of DPSCs make them a subject of great interest in biomaterials research [9,10].

Biomaterials, including a broad range of polymers, are engineered to interact safely and beneficially with biological systems and are pivotal in numerous biomedical applications such as dental and bone tissue regeneration [11,12]. Loosening or poor bone remodeling of implants may occur, making it important to search for materials that allow the preparation of structures even more similar to the natural bone matrix, with polymers being excellent candidates for this application [13]. Selecting the appropriate polymer for therapeutic use with DPSCs is an intricate process, requiring the material to be analogous to the physiological and biochemical properties of the tissue it is intended to replace and, importantly, to exhibit high biocompatibility. However, this selection process presents challenges, like the body’s potential adverse responses and managing the suitable degradation rates. Material science, technology, and biology advancements promise to bring forth increasingly effective and innovative polymers [14]. Hence, the synergy between DPSCs and polymers holds immense potential, opening new roads in regenerative medicine and tissue engineering. Indeed, DPSCs were shown to be biocompatible with various polymers such as Chitosan polymeric blends of polybutadiene (PB), poly-ε-caprolactone (PCL)/poly (rotaxane), poly(lactide-co-caprolactone) (PLCL), and several others [15,16,17,18]. Polylactic Acid (PLA) and Acrylonitrile–Butadiene–Styrene (ABS) are promising in this broad spectrum of polymers, each with their unique characteristics and potential advantages for use in conjunction with DPSCs.

PLA is a thermoplastic material derived from renewable resources such as cornstarch or sugarcane, making it biodegradable and environmentally friendly. This material typically requires a lower extrusion temperature of around 180–220 °C, making it easier to work with, particularly for beginners or those with lower-end 3D printers [19,20,21]. PLA-based formulations, which are FDA-approved for numerous applications, offer an advantage in terms of speedy clinical implementation. Despite its extensive use in surgeries, implants, and drug deliveries, PLA confronts several physical and biological hurdles, such as limited solubility in water, poor retention at the administration site, inadequate bioavailability, and issues with longevity. To counteract these limitations, PLA is often combined with different biocompatible polymers to create nano formulations for medical therapeutic deliveries, a process also sanctioned by the FDA. The human body can degrade and metabolize PLA into LA, affirming its status as an FDA-approved material for biomedical purposes. These multiple FDA approvals validate the extensive adaptability of PLA in clinical settings [22,23]. However, other polymers used for 3D printing may overcome several limitations of PLA, one of which might be ABS. 

ABS is a petroleum-based thermoplastic, which is non-renewable and not biodegradable. This material requires a higher extrusion temperature of about 210–250 °C, which can make it a bit more challenging to work with. However, ABS is generally stronger, more durable, and more flexible than PLA, making it a preferred choice for parts that need to resist wear and tear. It has a matte finish, and its components can be easily sanded and treated with acetone to create a smooth surface [24]. However, ABS is not as environmentally friendly as PLA, as it is not biodegradable, and its production and disposal have a higher environmental impact. It requires a heated print bed and, ideally, an enclosed print chamber to control cooling as it tends to warp or shrink during the process. While using ABS, adequate ventilation is necessary as it produces a strong, unpleasant smell and emits styrene, a potential carcinogen, when heated. The cost of ABS is generally comparable to PLA, although this may also vary [25,26].

Although it is not as strong or durable as ABS, PLA is quite sturdy and highly stiff, which can be beneficial in certain use cases. The PLA post-processing capabilities allow it to be sanded and primed for painting; however, it does not respond well to acetone for smoothing. It can be printed without a heated bed or in an open printer environment, as it does not warp as easily as ABS. It is generally considered safer to use and emits a sweet smell when heated. While PLA is environmentally friendly and derived from renewable resources, its production process can be more complex and costly than ABS, a petroleum-based plastic. However, despite PLA’s higher cost, its biodegradability and biocompatibility often make it a preferred choice for certain biomedical applications over ABS [27,28]. The choice between the two polymers depends on the balance between cost and the required properties for the specific application. Furthermore, little is known about PLA interaction with MSCs, including DPSCs. Moreover, ABS is not well understood in the context of various cell types.

Considering the gaps in knowledge, our study aims to elucidate the interaction of DPSCs with ABS and PLA. Both ABS and PLA have attractive physical–mechanical properties for tissue engineering applications through the production of scaffolds in 3D printers. Due to their optimal properties for tissue engineering, these polymers are ideal for 3D printing, and offer the necessary chemical, mechanical, and flexibility characteristics for creating complex scaffolds. This makes them particularly suitable for fabricating implantable medical devices and structures for bone tissue replacement, which may include interconnected pores or reinforcements like hydroxyapatite and graphene nanoplatelets. Their effectiveness in these applications is supported by various studies [29,30,31,32]. In this comprehensive examination, we assessed several crucial aspects related to the behavior of DPSCs when they come into contact with ABS and PLA. These parameters included the adhesion and proliferation of DPSCs on the polymer surfaces, their cytotoxic response, the preservation of their genomic stability, and their capacity for osteogenic differentiation. Furthermore, the stability of the immunophenotypic profile and the immunogenic characteristics of the DPSCs in the presence of these polymers were also investigated. This multifaceted analysis aimed to provide a detailed and coherent understanding of the biocompatibility and potential impacts of ABS and PLA polymers on DPSCs.

## 2. Materials and Methods

To evaluate the biocompatibility of DPSC with ABS and PLA polymers, three study groups were defined. DPSC corresponds to the control, where DPSCs are plated on usual surfaces for cell culture, such as polystyrene microplates. DPSC+ABS: corresponds to the DPSC plated on the ABS scaffold. DPSC+PLA: DPSC cultured on the PLA scaffold.

The Local Ethics Committee approved this study (approval number: 1.838.022).

### 2.1. DPSCs Isolation and Culture

The dental pulp from three healthy donors of both sexes, age 15–24 years, was mechanically removed with the help of an endodontic file (Hedstroem—type H) and then fragmented with the aid of a scalpel blade. The cell suspension was dissociated by the action of type II collagenase enzyme (0.0048 g/mL) (Gibco^TM^, Grand Island, NY, USA), under agitation, at 37 °C, for 1 h, filtered (40 µm), and centrifuged (10 min at 453× *g*) in 20 mL of phosphate buffered saline (PBS) (Gibco^TM^, Grand Island, NY, USA). Subsequently, the cells were plated in a flask with a growth area of 25 cm^2^ with a volume of 5 mL of medium composed of Iscove Modified Dulbecco Media (IMDM) (Gibco^TM^, Grand Island, NY, USA), supplemented with 20% fetal bovine serum (FBS) (Gibco^TM^, Paisley, UK) and 1% antibiotic (500 U of penicillin and 500 µg of streptomycin) (Gibco^TM^, Grand Island, NY, USA), kept in an incubator at 37 °C with a 5% concentration of CO_2_ and 95% humidity. The culture medium was replaced three times a week.

When the cultures reached approximately 80–90% confluence, enzymatic dissociation was performed using trypsin/EDTA (0.25%) (Gibco^TM^, Grand Island, NY, USA). Each time this procedure is performed, it counts as one passage (P). All experiments were performed between the third (P3) and fifth passage (P5) of the cells. In the analyses by flow cytometry, bioluminescence quantification, and genetic stability analysis, the “Control” samples were plated in culture flasks or plates. In cellular differentiation analyses and scanning electron microscopy, the “Control” samples were plated on glass slides due to the necessary manipulation to develop these techniques.

### 2.2. Cell Transduction

HEK293 cell line was transfected with the vectors pMD2.G, pCMV_dr8.91, and pMSCV_Luc2_T2A using Lipofectamine 2000 (Invitrogen, Waltham, MA, USA) and maintained in culture for three days with Modified Eagle Medium (DMEM) (Gibco^TM^, Grand Island, NY, USA), supplemented with 10% FBS (Gibco^TM^, Paisley, UK) and 1% antibiotic (Gibco^TM^, Grand Island, NY, USA) in an incubator at 37 °C with a 5% concentration of CO_2_ and 95% humidity. The supernatant containing viral particles was collected, filtered (0.22 µm), and ultracentrifuged at 100,000× *g* for 1 h and 30 min. The pellet was resuspended in 1% PBS/BSA (Sigma-Aldrich, St. Louis, MO, USA), distributed in 40 µL aliquots and stored at −80 °C. DPSCs cultured under the conditions described previously were then transduced with the viral particles and 10 µg/µL of hexadimethrine bromide (Polybrene, Sigma-Aldrich, USA). The medium used for cell transduction was changed every 24 h for three days. After this period, 10 mM of puromycin (Sigma-Aldrich, St. Louis, MO, USA) was added to the cultures for the selection of the transduced cells. 

A week after the introduction of the lentiviral vector into the cells, they were evaluated for bioluminescence. A total of 30,000 cells (told previously in Neubauer’s chamber) were plated in a well of a 12-well plate, and the emission of a light signal was measured after supplementing the culture medium with D-luciferin (150 µg/mL) (Perkin Elmer, Waltham, MA, USA).

### 2.3. Preparation of 3D-Printed Scaffolds

The polymers analyzed for biocompatibility for use as biomaterials were ABS (ABS F3DB™, Novo Hamburgo, Brazil) and PLA (PLA Cubex™, Tokio, Japan). The tested specimens were prototyped using a 3D printer (VOID3D^®^, Natal, Brazil) with an extrusion temperature of 190 °C and a brass printing nozzle with a 0.4 mm opening. The printing was carried out in a circular format in two sizes: 1.8 cm and 5.7 cm in diameter by 2 mm in thickness, layer by layer, and with orthogonal orientation. Following the prototyping phase, the polymers were washed with Extran^®^ MA 4% detergent (Merck S.A., Rio de Janeiro, Brazil) to remove any potential contaminants that could adversely affect cell cultivation. The specimens were left to dry at room temperature. After drying, each polymer sample was individually wrapped in surgical-grade paper and subsequently sterilized using ethylene oxide.

### 2.4. Analysis of Cell Adhesion and Proliferation

#### 2.4.1. Bioluminescence

Transduced DPSCs were analyzed for cell adhesion and proliferation. For this, 30,000 cells were plated in 100 µL of IMDM (Gibco^TM^, Grand Island, NY, USA) in the central region of polymer discs (1.8 cm in diameter) and on the control plate. The plates were placed in an incubator (37 °C) for 40 min for cell adhesion, then the volume was completed to 1.5 mL. After 24 h, the polymers were subjected to two washes with 2 mL of PBS (Gibco^TM^, Grand Island, NY, USA) and then transferred with the aid of sterile forceps to a new culture plate in order to prevent non-adherent cells or cells from adhering to the well of the plate from being quantified. 

The analyses were carried out by using the IVIS Lumina II device (Xenogen-Caliper, Perkin Elmer, Hopkinton, MA, USA), which is an optical image capture system sensitive to bioluminescence. The bioluminescent signal can be quantified. For this, a region of interest for analysis is determined, which corresponds to the area (cm^2^) that the signal occupies in relation to the emission of photons per second per steradian since the light signal is emitted in all directions. Only living cells degrade the substrate, generating the light signal. The image generated in the analysis demonstrates the dispersion of cells over the evaluated surface. The progressive increase in the bioluminescence signal captured over the days analyzed allows us to confirm cell proliferation. For each analysis, 150 µg/mL of D-Luciferin (Perkin Elmer, Waltham, MA, USA) was added to the DPSCs, and a consecutive series of images was captured with 30 s of exposure until the highest intensity of light signal emission was obtained, which was used in the comparative analyses between the 2nd and 7th day of follow-up after the DPSCs plating. The Living Image software version 4.1 was used for the analyses. 

#### 2.4.2. Scanning Electron Microscopy

Scanning electron microscopy (SEM) allows analyzing cell morphology and adhesion and the microstructural characteristics of the polymer surface. The DPSCs were observed after 24 h, 7, 14, 21, and 28 days of cultivation on the polymers. For each moment of analysis, the samples composed of DPSC, DPSC+ABS, and DPSC+PLA were fixed in 1.5 mL of glutaraldehyde solution (Electron Microscopy Sciences, Hatfield, PA, USA) in sodium cacodylate buffer (Sigma-Aldrich, St. Louis, MO, USA) and sucrose (Biotec-Labmaster, Pinhais, Brazil)) for 45 min. The fixing solution was removed, and the sample was washed with sodium cacodylate buffer and sucrose for 10 min. After fixation, the samples were dehydrated in a series of 10-min incubations with ethanol solutions (35%, 50%, 70%, 100%) (Biotec-Labmaster, Pinhais, Brazil) followed by Hexamethyldisilazane (HMDS) (Sigma-Aldrich, St. Louis, MO, USA) 100% for the same time. After drying, the metallization in gold was performed (QUORUM—Q150R ES), where they were covered with a thin gold deposition for SEM analysis (SEM Vega3, Tescan, Brno, Czech Republic).

### 2.5. Genetic Stability

In order to verify whether the polymers can cause any chromosomal changes to the karyotype of DPSCs, analyses were conducted at two points in time: day 0 and after 14 days of cell culture with conditioned medium by ABS and PLA polymers. This analysis technique requires visualization of cell proliferation under a microscope, which is not possible due to the opacity of the polymers. Therefore, for this specific analysis, it was decided not to culture the cells on the polymers and to use a culture medium conditioned by the polymers. The use of this medium is described in the polymer cytotoxicity tests for “Biological Evaluation of Medical Devices, part 5—Tests for cytotoxicity: in vitro methods” according to the International Standardization Organization (ISO) 10993-5 (2009) standard. 

The media were individually prepared with each polymer, ABS, and PLA. For each 1 mL of culture medium IMDM (Gibco^TM^, Grand Island, NY, USA), 1.25 cm^2^ of granulated polymer was used. The polymers were previously washed and sterilized by ethylene oxide in the same way as the disks used in the other analyses. The medium was in contact with the polymer granules for 48 h at 37 °C, after which it was filtered (0.22 µm), the polymers discarded, and the medium was supplemented with 1% antibiotic (Gibco^TM^, Grand Island, NY, USA) and 20% FBS (Gibco^TM^, Paisley, UK). During the 14 days of cultivation, three medium changes were performed per week.

The protocol of Borgonovo et al. (2014) was used to analyze genetic stability. In this procedure, 1 × 10^6^ cells were plated in flasks with a growth area of 150 cm^2^. After 48 h 0.1 µg/mL of colchicine (Sigma-Aldrich, St. Louis, MO, USA) was added to the culture to halt mitosis. The DPSCs were monitored under an inverted microscope for three-to-six hours to analyze the ideal point for harvest for cytogenetic tests. The sample underwent enzymatic dissociation with trypsin/EDTA (0.25%) (Gibco^TM^, Grand Island, NY, USA). Exposed to potassium chloride (KCl) (0.057 M) (Sigma-Aldrich, St. Louis, MO, USA) for 30 min at 37 °C. The sample fixation was carried out by the addition of fixative 3:1 (methanol/acetic acid) (Merck, Darmstadt, Germany) twice, centrifuged for 8 min at 400× *g*, and the supernatant discarded. A new fixative solution was added in the proportion 2:1 (methanol/acetic acid) (Merck, Germany), centrifuged (8 min at 400× *g*), repeating twice. The sample was then stored at −20 °C. 

For the preparation of slides and banding, drops of the sample were added onto the slides in a humid environment and heated to 60 °C to evaporate the fixative. The sample was then dehydrated in a 60 °C oven for a period of 12 h. The slides were immersed in the following sequence of solutions: trypsin 1:250 (Gibco^TM^, Grand Island, NY, USA), FBS/NaCl solution (1:40) (Gibco^TM^, Paisley, UK /Sigma-Aldrich, USA), distilled water, Giemsa stain (Laborclin, Brazil). The slides dried at room temperature. On average, 20 metaphases were photographed, and the karyograms were assembled and analyzed with the help of the LUCIA software (Laboratory Universal Computer Image Analysis, from LIM—Laboratory Imaging s.r.o.) (Laboratory Imaging, Prague, Czech Republic).

### 2.6. Analysis of the Potential for Osteogenic Cell Differentiation

For future clinical applications of ABS and PLA polymers, DPSCs were plated on the polymers and stimulated for differentiation in the osteogenic lineage. The polymer discs (5.7 cm) were placed on a Petri dish, and 100,000 cells were plated for the evaluation of the CD105 and osteocalcin markers by flow cytometry (protocol described later). Also, 20,000 cells were plated on glass coverslips and polymer discs (diameter of 1.8 cm) for differentiation evaluation by cytochemistry. After verification of cell confluence by optical microscopy, the culture medium was replaced by the commercial osteogenic differentiation induction medium (Lonza, Walkersville, MD, USA), which contains dexamethasone, L-glutamine, penicillin, streptomycin, MCGS (mesenchymal cell growth supplement), and β-glycerophosphate. Medium changes were made three times a week for three weeks. At the end of the differentiation period, the DPSCS were dissociated from the plate and polymers, requiring an 8-min incubation with the trypsin enzyme (Gibco^TM^, Grand Island, NY, USA), and analyzed by the flow cytometry technique for the immunophenotypic profile to show if there is an expression of a characteristic marker of the osteogenic lineage. The cells grown and differentiated on the coverslips were washed with 2 mL of PBS (Gibco^TM^, Grand Island, NY, USA), fixed in 2 mL of paraformaldehyde (Sigma-Aldrich, St. Louis, MO, USA), washed again with 2 mL of deionized water, incubated with 2 mL of Alizarin Red S (Sigma-Aldrich, USA), then washed with 2 mL of deionized water and dehydrated using the sequence of acetone, acetone + xylene, xylene (Sigma-Aldrich, St. Louis, MO, USA), then the slides were mounted with resin. Alizarin Red S allows visualization of the extracellular matrix rich in calcium. Scanning Electron Microscopy (SEM) was performed to analyze the morphology of DPSCs using the fixation, dehydration, and metallization protocol described earlier (2.4.2). Osteogenic differentiation was quantified following the protocol described in Utumi et al. 2021 [33]. The samples were washed with PBS (Gibco^TM^, Grand Island, NY, USA) and distilled water and fixed with 100% ethanol (Merck, Germany) for 15 min. The Alizarin Red S dye was later added for 40 min. It was washed again with PBS (Gibco^TM^, Grand Island, NY, USA) and distilled water, and a solution of 10% acetic acid (Sigma-Aldrich, St. Louis, MO, USA) and 20% methanol (Darmstadt, Merck, Germany) was added for 15 min in an orbital shaker at room temperature (approximately 21 °C). Absorbance was quantified by spectrophotometry (VersaMax Microplate Reader^TM^, Molecular Device^TM^, San Jose, CA, USA) at a wavelength of 450 nm. The test was performed in triplicate. As white standard, a solution of 10% acetic acid and 20% methanol was used.

### 2.7. Immunophenotypic Evaluation of the Cells

The characterization of the cells was carried out before (day 0) and after (14 and 28 days) the cultivation of the cells on the polymers with specific antibodies for MSC based on the definition of Dominici and colleagues (2006). In this definition, the cells must exhibit high expression of the markers CD29, CD73, CD90, and CD105 and low expression of the markers CD14, CD19, CD34, CD45, and HLA-DR. For cell viability analysis, the 7AAD dye was used, and to evaluate apoptosis, annexin V was used. MSC presents immunomodulatory properties and low immunogenicity, minimizing the possibility of immune rejection against the transplanted cells. The evaluation of the expression of HLA antigens (HLA-ABC (class I) and HLA-DR (Class II)) and costimulatory molecules (CD40, CD80, and CD86) was carried out in order to check whether the cultivation of DPSCs on the polymers alters the expression of these markers, which may alter the inhibition profile of the immune system’s response. For osteogenic differentiation analysis, the markers CD105, which identifies immature and undifferentiated cells, and osteocalcin, which is expressed by osteoblasts, were used. This analysis was carried out at two times: day 0 and 21 days after the cultivation of the cells on the polymers. The data obtained were analyzed in the FlowJo software (FlowJo^®^) version 10.

All the antibodies used in the study were manufactured by BD Pharmigen (San Diego, CA, USA). A total of 100,000 cells were used, distributed over 5 polymers in the form of a disc (1.8 cm in diameter) to obtain the necessary number of cells for cytometry. The cells were plated in a volume of 250 µL of IMDM (Gibco^TM^, Grand Island, NY, USA) culture medium supplemented with 1% antibiotic (Gibco^TM^, Grand Island, NY, USA) and 20% FBS (Gibco^TM^, Paisley, UK) on the polymers. The plates were placed in an incubator (37 °C) for 40 min for cell adhesion, then the volume was completed to 1.5 mL. The medium change was performed three times a week. After the cultivation period, the cells were dissociated from the polymers with the enzyme trypsin/EDTA (Gibco^TM^, Grand Island, NY, USA) as previously described.

After enzymatic dissociation, the cells were distributed at a minimum density of 400,000 cells per tube. A 500 µL volume of PBS (Gibco^TM^, Grand Island, NY, USA) was added and then centrifuged at 800× *g* for 5 min. The supernatant was discarded, and markers were added according to the analysis. The cells were then incubated for 30 min at room temperature in the absence of light. Subsequently, 500 µL of PBS (Gibco^TM^, Grand Island, NY, USA) was added, followed by another round of centrifugation at 800× *g* for 10 min. The supernatant was again discarded, and the cells were reconstituted and fixed in 500 µL of 1% paraformaldehyde, with the exception of the tube containing the 7-AAD dye, which was reconstituted only in PBS (Gibco^TM^, Grand Island, NY, USA).

For the tube where the osteocalcin marker was analyzed, before fixation, additional steps were taken to permeabilize the cells, as this marker is intracellular. A 100 µL volume of reagent A from the “Fix and Perm^®^” kit (Invitrogen, Carlsbad, CA, USA) and 100 µL of the sample were added, followed by incubation at room temperature, in the dark, for 15 min. Then, 2 mL of PBS (Gibco^TM^, Grand Island, NY, USA) was added, and the sample was centrifuged for 5 min at 40× *g*. The supernatant was discarded, and the monoclonal osteocalcin antibody and 100 µL of reagent B from the “Fix and Perm^®^” kit were added. The sample was incubated at room temperature, in the dark, for an additional 20 min. Then, 500 µL of PBS (Gibco^TM^, Grand Island, NY, USA) was added, and the sample was centrifuged for 5 min at 400× *g*. The supernatant was discarded, and the cells were reconstituted and fixed in 500 µL of PBS (Gibco^TM^, Grand Island, NY, USA) with 1% formaldehyde (Merck, Damstadt, Germany). 

The cell analysis was conducted using a FACSVerse flow cytometer (Becton and Dickinson^®^, Macquarie Park, NSW, Australia), which allows up to eight parameters to be analyzed per sample. A total of 100,000 events were collected from each sample. The equipment adjustment was performed for the conditions of cell size and complexity analysis. The adjustment for fluorescence parameters was performed by incubating the cells with isotype controls, chosen according to the markers used in each analysis. The reduction of interference between fluorophores in different fluorescence channels was accomplished by compensation. The obtained data were analyzed in the FlowJo software (FlowJo^®^) version 10. All the antibodies used in the study were manufactured by BD Pharmigen.

### 2.8. Statistical Analysis

The experiments were conducted in technical and biological triplicates, and the results were expressed as means and medians, with the results presented in tables and graphs. Osteogenic differentiation was performed in biological duplicate, and the results are descriptive. Statistical analyses were performed using the Statistical Package for the Social Sciences (SPSS) software version 25. Values of *p* < 0.05 were considered significant.

### 2.9. Proteomic Analysis of the DPSCs (Secretome)

Conditioned medium from 1.0 × 10^5^ cells plated over 5.7 cm diameter disks (groups DPSC, DPSC+ABS, and DPSC+PLA) was obtained after 48 h in culture without FBS. The conditioned medium (3 mL) was then concentrated in Amicon 3 kDa NMWCO (Amicon^®^ Ultra-15, Millipore Merck, Damstadt, Germany) and quantified using a Qubit 2.0 fluorometer (Thermo Fisher Scientific, Waltham, MA, USA). Approximately 20 μg of culture medium from cells were processed as previously described by Angulski et al. (2017). Briefly, lysis buffer (100 mM Tris-HCl, pH 7.5, 4% SDS, 100 mM DTT, and H_2_O 18.2 MΩ.cm) was added to the concentrated CM at a ratio of 1:1 (*v*/*v*), 15 min at 94 °C. Samples were sonicated for 30 min, centrifugated at 16,000× *g* for 5 min, and separated by 10% SDS-PAGE. The proteins were subjected to in-gel tryptic digestion. After protein reduction (10 mM DTT in 50 mM ABC) and alkylation (55 mM iodoacetamide in 50 mM ABC), gel pieces were dried and rehydrated in trypsin solution (trypsin 12.5 ng/μL in 50 mM ABC) and incubated overnight at 37 °C. After, peptides were extracted using an extraction buffer (3% trifluoroacetic acid (TFA) and 30% acetonitrile (MeCN) and desalted in C18 spin columns. The peptides were eluted slowly in 80% MeCN, 0.1% formic acid, resuspended in 0.1% formic acid, and 5% DMSO (Sigma-Aldrich, St. Louis, MO, USA) and analyzed by LC-MS/MS. Proteomic analysis was carried out in biological triplicates per condition as previously described in detail by Angulski et al., 2017 [34]. To define the level of analysis, stringency was only considered for further analyses of the proteins with a minimum of two valid values in at least one group (Control, PLA, or ABS). IMDM (Gibco^TM^, Grand Island, NY, USA) without FBS was used as a negative control.

To obtain the Gene Ontology (GO) annotation of the biological processes, molecular functions, and cellular components of the identified proteins, an enrichment analysis of the gene sets was performed using Funrich 3.1.3 [35]. Differentially expressed protein (DEP) analysis was performed with Perseus software version 2.0.10.0 [36] using ANOVA multi-sample test. The fold change of the DEPs was calculated as the ratio of the normalized LFQ values. If necessary, the intensity value of proteins absent in any sample was replaced by an arbitrary background value of 1.00 E06. 

## 3. Results

### 3.1. Collection, Isolation and Maintenance of DPSCs

Samples from the pulp of permanent teeth were successfully isolated from three donors. The cells adhered to the culture flask displayed fibroblastoid morphology and took an average of 17 days to reach 80% confluence after isolation (Appendix A). Passage 1 was performed at this time (Appendix A). No contaminations or other changes in cell culture were observed. 

### 3.2. Adhesion, Proliferation, and Cell Morphology of DPSCs Cultivated on the Three Surfaces

After the introduction of the lentiviral vector into the DPSCs, they were evaluated in vitro, confirming their transduction by emitting a luminescent signal after supplementation of the culture medium with D-luciferin (Figure 1A). The images obtained from the evaluation of the bioluminescence emitted by the DPSCs over the days of culture follow-up the verification of cell viability, adhesion, localization, how dispersion occurred, and the occurrence of cell proliferation on the analyzed surfaces: culture plate and scaffolds ABS and PLA (Figure 1B).

The bioluminescent signal emitted by DPSCs in both conditions and moments was equated on the same scale and then quantified. The values obtained from the average analysis of each condition were compared in the graph in Figure 1C. The highest intensity of the bioluminescent signal was emitted by the group DPSC, followed by group DPSC+ABS and DPSC+PLA. Statistical analysis of the data shows that there was no significant difference between the groups analyzed each day (Appendix A). Intragroup comparison at different times also showed no significant difference (Appendix A). Although the image Figure 1B obtained in the analysis of the bioluminescent signal emission shows a discrepancy between the analyzed groups, the statistical analysis of the data proves that there was no significant difference when analyzing the cultures. The significant difference is only found when the analysis occurs among the days analyzed within each group (*p* < 0.001 for each individual group).

SEM allowed us to analyze DPSCs and verify the surface topography of both groups (Figure 2). The prototyping of the polymers was designed to be printed in orthogonal layers between which cracks could be observed. The filaments presented a convex surface with little irregularity. The DPSCs were anchored to the surfaces of the polymers, and their initial morphology was elongated, typical of MSCs. With seven days of culture, in addition to the cell anchoring, it was possible to visualize filopodia that adhered between the surface irregularities.

After 14 days of culture, it was no longer possible to visually determine the cellular contours, and the cracks between the polymer filament wefts were covered by cells in the group DPSC+ABS and DPSC+PLA. The cells, which were initially adhered in a dispersed manner, by the end of the follow-up (28 days) had proliferated, forming a tissue that displays more condensed cellular and extracellular matrix arrangements. It was found that there was better interaction between the DPSCs and the studied polymers than with the glass surface used as control. During the cell culture of DPSCs on the slides, many cells detached and were discarded during medium changes, which might explain why SEM shows fewer cells in individualized form and elongated morphology, with no formation of a structured tissue. 

### 3.3. Genetic Stability of DPSCs Verified by Giemsa Banding Chromosomal Analysis

Previous and subsequent analysis, 14 days of cell culture with medium conditioned by ABS and PLA polymers, of the genetic stability of DPSC demonstrated a normal karyotype, with metaphases without clonal chromosomal abnormalities (Figure 3).

### 3.4. Analysis of Osteogenic Differentiation Potential

For the analysis of cell differentiation in the osteogenic lineage, samples from two DPSCs donors were plated on glass slides or on polymers (1.8 cm diameter). The culture on the glass slide allowed for follow-up via optical microscopy of cell proliferation and differentiation. This culture was photographed weekly for a period of 21 days, which allowed us to follow the modifications in cell morphology (Appendix A). Osteogenic differentiation was confirmed by Alizarin Red S staining that highlights in red the deposition of calcium in the extracellular matrix, which was not observed in non-stimulated cells (Figure 4A). The differentiated DPSCs were also observed by SEM, where it was possible to visualize a structured tissue with nodules related to calcium deposition. This was more evident in the samples cultivated on the polymers than on the slide (Figure 4B). Still using staining, osteogenic quantification was carried out, where we found greater optical density in samples stimulated to differentiate when compared to their controls, with the data confirmed by statistical analysis (Figure 4C).

For the characterization of DPSCs after osteogenic differentiation stimulus, the cells were detached from the different surfaces (with 5.7 cm diameter) with the aid of the trypsin enzyme and marked with osteocalcin, CD105. The immunophenotypic profile showed an increase in osteocalcin expression and a decrease in CD105 expression, as well as a decrease in viability after differentiation (Figure 4D).

### 3.5. Immunophenotypic Characterization of DPSCs

The cell characterization analysis presented an immunophenotypic profile compatible with the DPSCs profile before and after cell culture on polymers at 14 and 28 days, expressing CD29, CD73, CD90, CD105, CD166, and lack expression for CD14, CD19, CD34, CD45 (Table 1).

### 3.6. Cell Viability

Cell viability was high in all analyses. The results obtained in the analysis of group DPSC (day 0) by the 7-AAD dye showed 98.4% cell viability prior to culture on polymers. The results obtained after 14 days of culture on group DPSC+ABS were 96.6% and 96.2% on group DPSC+PLA. For the 28-day culture, cell viability was 88.6% in group DPSC+ABS and 88% in group DPSC+PLA. All results obtained in the analyses showed low expression of Annexin V (Table 2). On day 28, vigorous washing was required after enzymatic dissociation to remove cells from the polymers.

### 3.7. Expression of HLA Antigens and Costimulatory Molecules

The results of the immunogenicity of DPSCs before cultivation on polymers were positive for HLA-ABC and negative for HLA-DR, CD80, and CD86. This profile was maintained in evaluations at 14 and 28 days (Table 3).

### 3.8. Protein Content and GO Analysis Reveal That the Secretome of DPSC Cultivated on Polystyrene, PLA, and ABS Are Highly Similar

To determine the protein composition of the secretome of DPSCs grown on Polystyrene, PLA, and ABS, a proteomic analysis using nano-LC-MS/MS was performed. Samples obtained from three independently isolated DPSCs in each biomaterial were prepared. LC-MS/MS analyses were conducted for each sample in triplicate. Only proteins that had a False Discovery Rate (FDR) value ≤ 1% were considered for analysis. A total of 774 proteins were identified. Considering proteins that were detected in at least two replicates of at least one condition resulted in a list of 504 proteins (Support information Appendix A). The comparison between the identified proteins in each sample revealed that approximately 51% of the proteins identified (256) were shared between the three conditions. A small percentage of proteins were unique for each material: <1% for PLA, 7% for ABS, and 23% for the control (DPSC). A higher number of proteins was identified in group DPSC (457) in comparison with group DPSC+ABS (362) and group DPSC+PLA (289). These data demonstrate that, at least from the point of view of protein content, the secretome from DPSC cultivated in the three different materials are, for the most part, qualitatively similar (Figure 5A).

To investigate the potential functions performed by the proteins present in the secretomes, we conducted functional enrichment analyses using Funrich software 3.1.3 [35]. The GO analyses of the proteins showed a high degree of similarity between the enriched GO terms found in the secretomes derived from DPSC in different conditions. Among the most significant GO terms for cellular components of the proteome analyzed are exosomes, extracellular space, and collagen-containing extracellular matrix (Figure 5B). The GO cellular component terms denoted the origin and quality of the samples analyzed. The GO results reveal the fact that all samples shared much of their protein contents. GO terms such as negative regulation of apoptotic processes are among the 15 most significant terms in PLA and control but not in ABS. 

Because our approach employed a non-saturating proteomic analysis, it was undesirable to focus solely on the sets of proteins found exclusively in one given condition. Also, the number of proteins exclusively found in each condition was low.

#### Only Eight Proteins Were Differentially Overrepresented in the Secretome of Any of the Conditions

To gain improved insight into the putative differences between the proteomic profiles of the secretome from the three conditions analyzed, a quantitative analysis was carried out using Perseus software version 2.0.10.0 [36]. Principal component analysis (PCA) and hierarchical cluster based on the normalized LFQ intensities of all the identified proteins produced similar results, showing that the samples do not segregate based either on the biomaterial or on the biological replicate, underlying the similarity already seen in the previous analyses (Figure 6A and Appendix A). The samples from both origins were qualitatively similar; when the abundance of the protein was considered, nine proteins were significantly overrepresented in any of the conditions with a *p*-value < 0.05 and a fold change ≥ 2.0 (Figure 6B,C). Since the protein of the SERPINA1 gene was present in the negative control (IMDM), it was not considered for further analysis. The genes whose proteins were overrepresented in the control condition in comparison with PLA and ABS were COL5A1, CLSTN1, FAM20C, SRPX, and HSPA13. The transforming growth factor-beta 2 (TGFB2) protein was overrepresented in the control and PLA relative to ABS. Higher levels of Vanin1 (VNN1) and COPA (COPI Coat Complex Subunit Alpha) proteins were overrepresented in PLA and ABS in comparison with the control condition (Figure 6C and Appendix A).

## 4. Discussion

Comprehending the interaction between ABS and MSCs, notably DPSCs, which exhibit high availability and differentiation potential, is of significant value. This is predicated on the fact that ABS displays several attributes that can surpass the well-documented limitations of other polymers like PLA. If ABS exhibits biocompatibility comparable to PLA, it could emerge as a viable alternative where the constraints of polymers such as PLA act as decisive limiting factors.

SEM analysis has shown that DPSCs exhibit a more flattened and elongated (fibroblastic) morphology at the beginning of cell culture. This same morphology was demonstrated at the start of cell culture in studies by Suarez-Franco et al. (2018) and Wang et al. (2010) when they analyzed MSC from the periodontal ligament on PLA and DPSC on PLLA [37,38]. At the time of plating, there is an initial adsorption of cells to the surfaces of the polymers, which shows favorable electrostatic and surface tension conditions, followed by adhesion/anchoring when the cells begin to secrete extracellular matrix [39]. After seven days of cultivation, DPSCs were observed in a three-dimensional format with cytoplasmic extensions and filopodia. Their anchoring occurred on irregular surfaces of the weave, promoting interaction between cells. These characteristics of the DPSCs were also observed in the study by Kwon et al. (2015) when they were plated on polyester [40].

In this study, we conducted an extensive analysis of dental pulp stem cells (DPSCs) interaction with two commonly used polymers for 3D printing—ABS and PLA. The results provide a comprehensive understanding of the impact of these materials on stem cell biology, which has implications for their use in biomedical applications personalized to the patient’s needs. Our analysis revealed that both ABS and PLA polymers supported the adhesion and proliferation of DPSCs. Importantly, neither ABS nor PLA polymers exhibited any significant cytotoxic effects that could compromise the viability of DPSCs. Suggesting that these materials could potentially serve as effective substrates for DPSCs growth. This is important, considering that stem cells often require highly specific conditions for growth and differentiation. This further establishes the potential biocompatibility of these polymers with DPSCs, making them promising materials for use in dental or orthopedic devices. 

The ability of PLA to support MSCs in vitro expansion was also previously confirmed by other studies. Serra et al. (2013) found that PLA and CaP glass scaffolds supported rat bone marrow mesenchymal stem cells (BM-MSCs) adhesion, suggesting their potential in tissue engineering [41]. Salerno et al. (2013) produced macroporous PLA scaffolds that enabled uniform cell adhesion, colonization, and proliferation of rat MSCs, promising for bone and cartilage regeneration [42]. Yagi et al. (2021) reported that honeycomb PLA films with 5 μm pores facilitated superior cell adhesion and cartilage formation using synovial MSCs, demonstrating their potential for in vitro cartilage engineering [43]. The adhesion, proliferation, and survival of MSCs on PLA surfaces were also previously observed in DPSCs [40,44,45] and human Periodontal Ligament Stem cells (PDLSC), which are also dental MSCs [37].

On the other hand, research examining the effects of cultivating cells on ABS has had controversial results in past studies. The study by Rosenzweig et al. (2015) presented comparable effectiveness of 3D-printed ABS and PLA thermoplastics as scaffolds for tissue engineering. Over three weeks, both supported cell proliferation, viability, and tissue generation. Nucleus Pulposus cells produced more matrix than chondrocytes on both materials, with no significant difference between the scaffolds [28]. Schmelzer et al. (2016) study analyzed the impact of four thermally printed materials—ABS, MED610, polycarbonate, and PLA—on primary human adult skin epidermal keratinocytes and BM-MSCs. Their results suggested potential toxic effects, particularly from MED610 and acrylonitrile–butadiene–styrene, which significantly affected both cell types, even without direct contact [46]. These findings underscore the importance of considering biological effects on specific cell types, in addition to mechanical properties, when selecting materials for regenerative medicine applications. 

In our study, we further evaluated the ability of DPSCs to differentiate osteogenically when cultivated on these polymer scaffolds. The results revealed that the DPSCs preserved their potential for differentiation, which suggests that these polymers might be suitable for applications in bone tissue engineering. Kolind et al. (2014) examined the influence of surface topography on osteogenic differentiation. The researchers found that altering pillar topography, particularly increasing the inter-pillar gap size from 1 to 6 μm for surfaces with smaller pillar sizes, not only led to decreased cell proliferation and more elongated cell structures with long pseudopodal protrusions. This also resulted in enhanced mineralization of DPSCs even when cultured without osteogenic differentiation factors in the medium. It was found that these topographical cues influenced cells even without additional differentiation factors, suggesting that surface design can play a critical role in controlling stem cell behavior [44]. Similarly, Kwon et al. (2015) highlighted the bone regeneration potential of hDPSCs. Utilizing a Poly-L-lactide (PLLA) scaffold, which is a specific form of PLA, the researchers were able to stimulate the hDPSCs to differentiate into osteoblasts. This led to the regeneration of more than 50% of new bone within cranial defects. The process was marked by the scaffold’s gradual degradation, indicative of successful neo-bone formation [40]. Moreno et al. (2023) developed unique scaffolds using PLA, wollastonite particles, and propolis extracts, exhibiting antibacterial activity against *Staphylococcus aureus* and *Staphylococcus epidermidis*, typical causes of osteomyelitis. Despite initial viability reduction, hDPSCs thrived by the seventh day. These results suggest that these novel scaffolds could be promising as bone substitutes capable of controlling typical severe infectious processes [45]. The possibility of prototyping with both ABS and PLA polymers allows the manufacture of scaffolds with complex geometries, both internal and external, which allows the production of infinite forms for patient care (Sears et al., 2016). Díaz and colleagues, 2017 highlighted the need to create scaffolds that have a network of interconnected pores in order to favor cell migration and the development of new bone tissue. Both ABS and PLA, when printed, can meet this demand.

The successful application of both polymers and cells goes beyond the delivery of the biomedical product, and biosafety is an obligatory step [47]. Two of the main biosafety concerns are related to the genetic stability of the cells and the potential immunoreaction of the cells under in vitro exposure and exposure to chemicals and culture surfaces [48]. McKee and Chaudhry (2017) reported the importance of elucidating cell signaling mechanisms in cultures on biomaterials and verifying gene stability. Therefore, we further investigated the chromosomal stability of DPSCs after contact with both tested polymers. Our results indicated genetic stability, as presented by the absence of clonal chromosomal abnormalities post-contact, which implies that these polymers do not induce genetic instability in DPSCs. This is supported by previous research supporting the genetic stability of DPSCs when cultivated under plastic culture flasks [49,50]. Finally, we analyzed the influence of these polymers on the immunophenotypic profile and immunogenicity characteristics of DPSCs. Our results indicated no substantial alterations in these properties, which supports the safe interaction between these polymers and DPSCs. These results represent crucial factors in ensuring the biosafety and efficacy of stem cell-based therapies. Thinking about the application of scaffolds produced in ABS and PLA and their incorporation into the body, in addition to a replacement graft, there is also the possibility of covering the prototypes with cells as these can help suppress the immune response of different cells, such as dendritic cells, T and B lymphocytes and natural killer cells [51,52].

The protein expression profiles of the DPSCs secretome displayed remarkable uniformity across Polystyrene, PLA, and ABS substrates. Of the numerous proteins evaluated, only eight exhibited significant differences in expression levels, suggesting a high degree of consistency in the protein secretome in both polymers. It is crucial to note that these few differences did not translate into observable functional changes in the other tests we performed in our study. Both ABS and PLA supported DPSC adhesion and proliferation of DPSCs without any noticeable cytotoxic effects. Moreover, both polymers preserved the osteogenic differentiation capabilities, genetic stability, and immunophenotypic profiles of DPSCs. In essence, the differential expression of these selected proteins did not appear to compromise DPSCs when cultivated on ABS and PLA. 

In accordance with our results, the maintained osteogenic differentiation capacity of DPSCs cultivated on PLA was verified by Alksne et al. (2022). They reported that rat DPSCs cultured on 3D-printed PLA scaffolds exhibited protein secretion patterns associated with bone and cartilage formation, even in the absence of chemical inducers for differentiation [53]. However, it is important to note that their study did not include a comparison with polystyrene surfaces, making it important to compare their results with ours. Furthermore, they have used non-human cells.

Our study provides valuable insights into the potential use of ABS and PLA polymers in biomedical applications involving DPSCs. However, further in vivo experimentation and extensive clinical studies are needed to confirm these findings and explore their potential therapeutic applications. PLA can be successfully combined with other biomaterials for various applications with MSCs. Zhou et al. (2016) showed a nanofiber film of polypyrrole and PLA could enhance neurogenic markers in umbilical-cord-derived MSCs. Raynald et al. (2019) demonstrated a PPy/PLA scaffold reduced scar tissue and promoted recovery in rat spinal cord injuries [54]. Lastly, Tambrchi et al. (2022) found that a Polycaprolactone and PLA scaffold could improve the differentiation of adipose-derived MSCs into cardiomyocytes [55]. These studies underscore the potential of PLA combined with MSCs in diverse biomedical scenarios. The same is valid for ABS, which, in our previous studies, presents similar biocompatibility features [46]. However, for ABS, more research is necessary because of the controversies regarding its effectiveness as a surface that allows the survival and proliferation of cells.

This comprehensive analysis has provided considerable insight into the interaction between DPSCs and two frequently employed polymers in 3D printing: Acrylonitrile–Butadiene–Styrene (ABS) and Polylactic Acid (PLA). ABS demonstrated biocompatibility comparable to the widely used PLA, a finding that expands our choices in the application of these polymers in cell therapy and regenerative medicine, particularly given the high proliferative capacity and easy accessibility of DPSCs. Despite these encouraging results, further research is needed to fully understand these interactions in more complex biological systems. Additional in vivo studies are crucial to verify these in vitro findings. Furthermore, it is essential to conduct comprehensive clinical studies. These studies will allow us to explore the potential therapeutic applications of ABS and PLA more deeply with DPSCs. 

## 5. Conclusions

This study emphasizes the importance of continued investigation into the use of ABS and PLA in biomedical applications involving DPSCs. Such endeavors could pave the way for significant advancements in the fields of stem cell biology and tissue engineering. Our findings suggest that ABS and PLA polymers not only foster the adhesion and proliferation of DPSCs but also support their osteogenic differentiation capabilities. Additionally, the results of our investigation confirm the genetic stability of DPSCs in contact with both polymers, a critical factor for the safety of stem-cell-based therapies. The absence of cytotoxic effects further emphasizes the potential biocompatibility of these materials since all analyses demonstrate similarities in results. The choice between the two polymers depends on the properties required at the time of the application of the scaffold. PLA is recommended for areas where the body can recover by replacing the scaffold with fabric and ABS when the area is very large and difficult to recover. The data suggest that both polymers are suitable for applications in bone tissue engineering and the creation of dental or orthopedic devices.

## Figures and Tables

**Figure 1 polymers-15-04629-f001:**
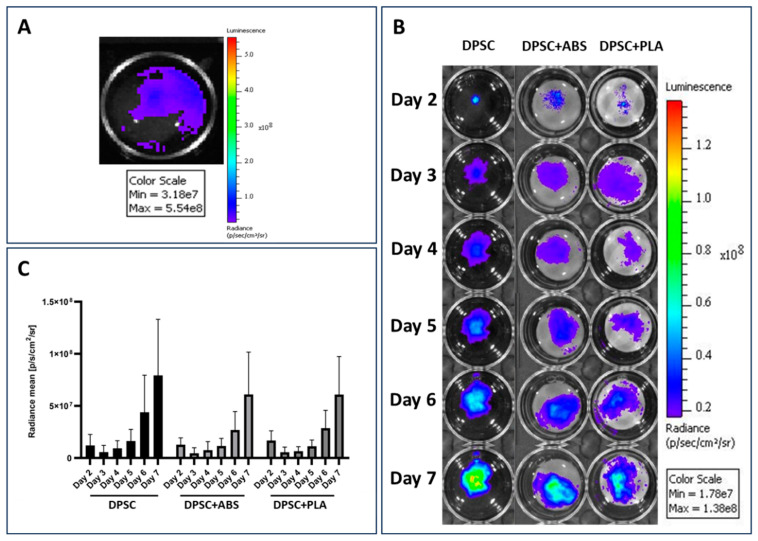
Adhesion and proliferation evaluation. (**A**) Confirmation of the expression of the luciferase enzyme by DPSCs. The light signal emitted by DPSCs during the degradation of D-luciferin by the luciferase enzyme present in the cells is represented by the color blue. (**B**) Evaluation of DPSCs cultured on the plate well (DPSC) and on ABS (DPSC+ABS) and PLA (DPSC+PLA) polymers. Representative pseudo-colored images of the bioluminescent signal emitted by cells. Monitoring was carried out between the 2nd and 7th day of cultivation. Comparison with the color scale allows us to confirm the increase in light signal emission over the days of cultivation, proving cell proliferation. For images (**A**,**B**): the side scale indicates the amount of light signal measured in the unit of measurement photons per second per square centimeter per steradian [p/s/cm^2^/sr]. (**C**) Quantification of the bioluminescent signal emitted by DPSC adhered to different growth surfaces. Unit of measurement photons per second per square centimeter per steradian [p/s/cm^2^/sr] (*n* = 9).

**Figure 2 polymers-15-04629-f002:**
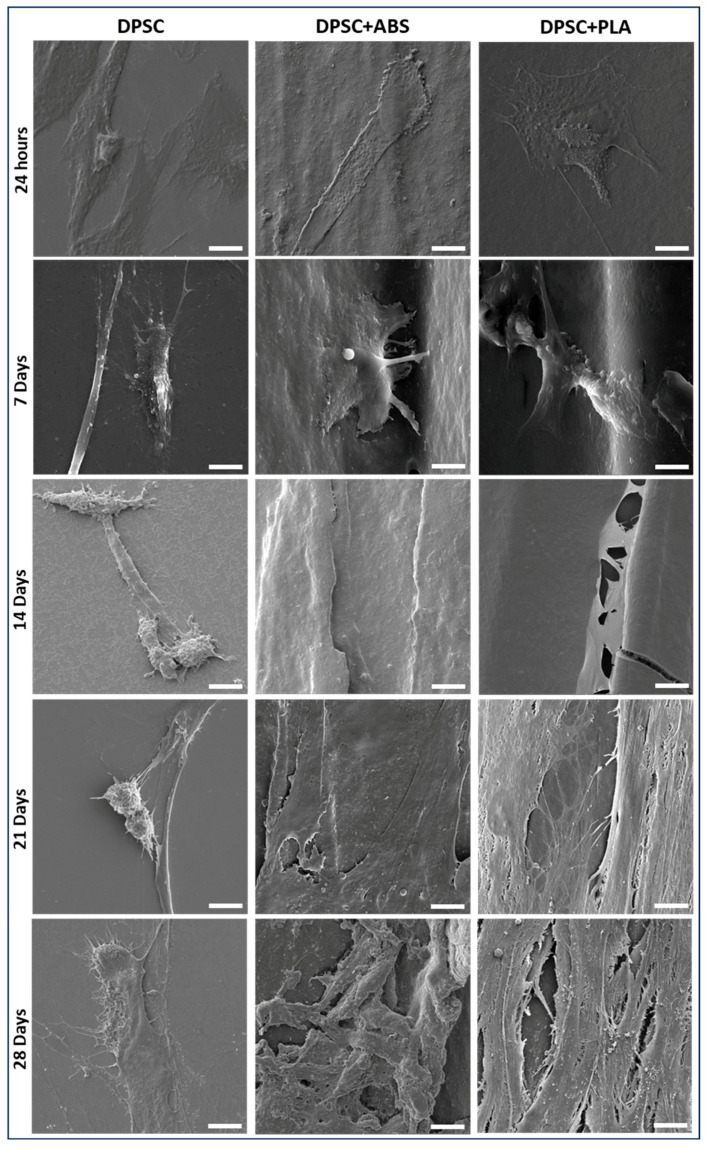
DPSC morphology for scanning Electron Microscopy (SEM) photomicrographs. Evaluation of the adherence of DPSCs cultured on glass coverslip (group DPSC), ABS (ABS F3DB^TM^, Brazil) (group 2), and PLA (PLA Cubex^TM^, Japan) (group 3), at the time intervals of 24 h, 7, 14, 21, and 28 days (magnification 3300×, scale bar: 10 μm).

**Figure 3 polymers-15-04629-f003:**
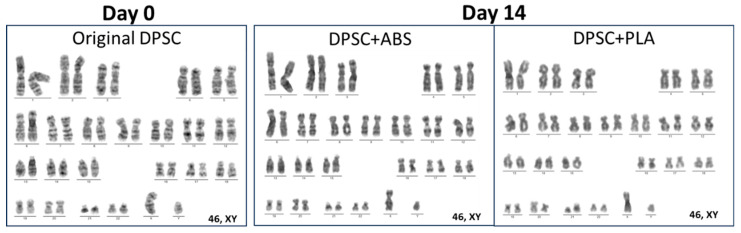
Representative karyograms: original DPSCs (day 0) and after 14 days of cultivation with medium conditioned in ABS (group DPSC+ABS) and PLA (group DPSC+PLA), presenting normal karyotypes under different time points.

**Figure 4 polymers-15-04629-f004:**
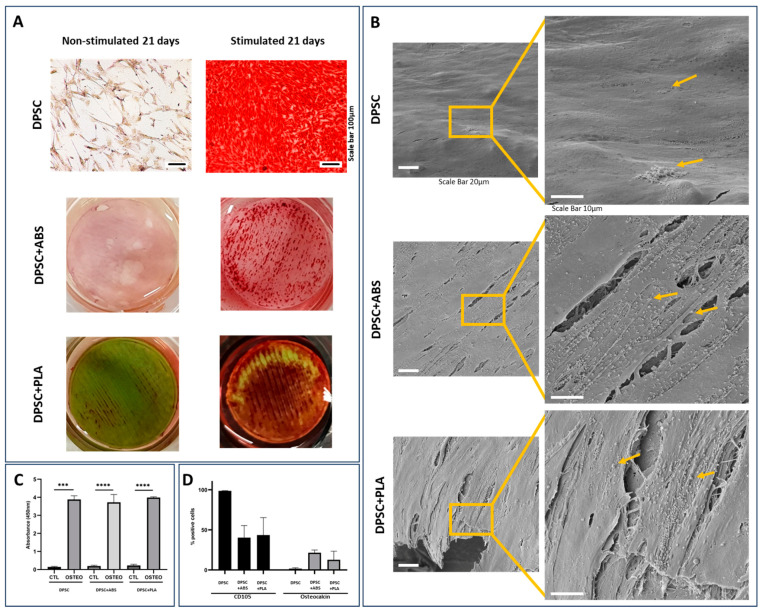
Analysis of osteogenic differentiation potential. (**A**) Alizarin S Red staining, 21-day cultivation. The red color shows calcium deposits in the extracellular matrix, proving osteogenic differentiation. DPSCs, when stimulated to osteogenic differentiation, demonstrate calcium deposition. DPSCs not stimulated to differentiate do not demonstrate calcium deposition. Photomicrograph of the culture on the coverslip with 100× magnification, scale bar: 100 µm. The polymers are 1.8 cm in diameter and were photographed. (**B**) Representative SEM photomicrographs of DPSC stimulated osteogenic differentiation for 21 days. High-magnification images show a structured tissue where it is possible to visualize cell filopodia, extracellular matrix, and nodules of mineral deposition indicated by arrows. Magnification 1400× (scale bar: 20 µm), 3300× (scale bar: 10 µm). (**C**) Osteogenic quantification comparing the groups analyzed with unstimulated DPSC (CTL) and stimulated (OSTEO) osteogenic differentiation ***: *p* = 0.0005, ****: *p* < 0.0001. (**D**) Immunophenotypic profile after osteogenic differentiation stimulus.

**Figure 5 polymers-15-04629-f005:**
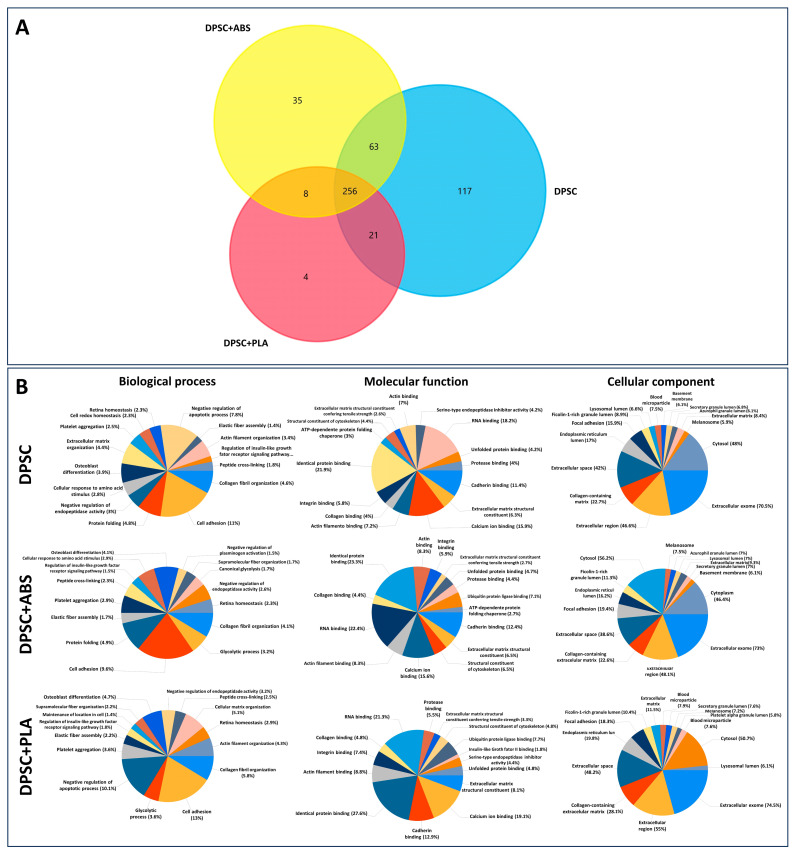
Protein identification and functional enrichment analysis of the secretome. (**A**) Venn diagram of the proteins identified in the three conditions. (**B**) Gene ontology enrichment analysis of total proteins identified in the secretome of cells cultivated on PLA, ABS, and control showing the most enriched terms for molecular function, biological process, and cellular components. The pie chart shows the 15 most significantly enriched terms in each category (*p* < 0.01). GO analysis was conducted with Funrich software (version 3.1.3).

**Figure 6 polymers-15-04629-f006:**
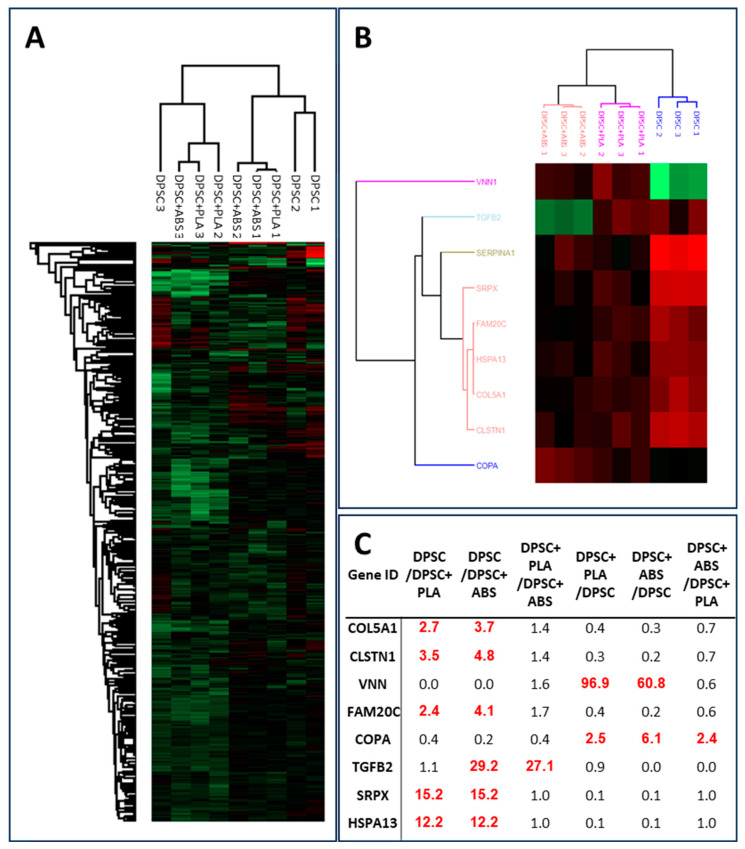
Eight proteins were differentially overrepresented in the secretome of any of the conditions. (**A**) Heatmap of the hierarchical cluster analysis based on the normalized LFQ intensities of all the identified proteins. (**B**) Clustering and heatmap visualization based on the ANOVA significant proteins. (**C**) Fold change of the eight differentially expressed genes comparing two by two the average of the normalized LFQ intensities of each tested biomaterial. Red numbers indicate a fold change > 2.00.

**Table 1 polymers-15-04629-t001:** Immunophenotypic profile of DPSC. Comparison of the data obtained in the characterization for mesenchymal stromal cell profile by flow cytometry of groups DPSC, DPSC+ABS, and DPSC+PLA at times of cultivation day 0, 14, and 28. The values are shown in percentage and refer to the average of three samples.

% Positive Cells	Group 1 (Day 0)	Group 1 (Day 28)	Group 2 (Day 14)	Group 2 (Day 28)	Group 3 (Day 14)	Group 3 (Day 28)
CD14	0.07	1.16	0.1	0.88	0.14	0.56
CD19	0.04	0.99	0.06	0.63	0.06	0.54
CD29	98.8	92.13	95.97	93.23	98.53	93.23
CD34	0.38	0.8	0.33	0.93	0.63	0.15
CD45	0.03	1.04	0.06	1.06	0.05	0.75
CD73	97.97	95.1	98.17	94.2	98.37	94.2
CD90	97.8	95.4	98.9	96.27	98.87	96.1
CD105	96.1	94.1	98.07	91.9	97.43	91.6
HLA-DR	0.31	0.92	1.02	1	1.8	0.88

**Table 2 polymers-15-04629-t002:** Analysis of viability and apoptosis. Profile of cell viability analysis (7AAD) and apoptosis indicator (Annexin V) of DPSC after cultivation on ABS and PLA polymers for 14 and 28 days. The values are shown as a percentage and refer to the average of three samples.

% Positive Cells	DPSC	DPSC+ABS		DPSC+PLA	
	Day 0	Day 14	Day 28	Day 14	Day 28
7AAD	1.59	3.39	11.44	3.95	11.93
Annexin V	0.18	1.48	2.96	1.48	1.57

**Table 3 polymers-15-04629-t003:** Expression of HLA antigens and costimulatory molecules. Comparison of the data obtained in the characterization for mesenchymal stromal cell profile by flow cytometry of groups DPSC, DPSC+ABS, and DPSC+PLA at times of cultivation day 0, 14, and 28. The values are shown in percentage and refer to the average of three samples.

% Positive Cells	DPSC		DPSC+ABS		DPSC+PLA	
	Day 0	Day 28	Day14	Day 28	Day 14	Day 28
CD40	0.003	3.39	2.76	2.63	2.58	2
CD80	0.009	0.88	2.43	1.36	2.54	0.45
CD86	0.02	11.64	6.20	5.16	6.50	5.59
HLA-ABC	89.57	96.07	93.80	96.33	90.77	96
HLA-DR	0.31	1.75	1.02	1.01	1.80	0.88

## Data Availability

Raw data, Proteins groups used for FunRich and Perseus analyses: https://docs.google.com/spreadsheets/d/1NUbN9lOu--r4hJGQ6JBfMw17W5OIK_6D/edit?usp=sharing&ouid=108025814496751279025&rtpof=true&sd=true.

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
