# Peer review of "Biocompatibility of ABS and PLA Polymers with Dental Pulp Stem Cells Enhance Their Potential Biomedical Applications"

_polymers, 2023, doi:10.3390/polym15244629_

Round 1

Reviewer 1 Report

Comments and Suggestions for Authors

The manuscript has been written well, and comments are given below to revise the manuscript –

1.      Abstract: It is well written, but two polymers are used - PLA & ABS. The authors have concluded that both are good, but I advise bringing a parameter to establish a better one in them.

2.      Introduction: A brief description of the other polymers reported is required to interact with DPSCs.  

3.      Line no. 139/ Is it exactly 30,000?

4.      2.2: Methods of culturing cells are missing.

5.      Figure 1B/ Lane-2: Better to write DPSC+ABS.

6.      Here, authors can compare both the polymers from the fl. intensity.

7.      3.3: Authors can explain Figure 1B.

8.      Discussion: 1st two paragraphs are very general, and that can be shifted to the introduction.

9.  Conclusion: Both polymers were given equal importance. A comparative statement will provide better insights.

10.  Decision: Minor revision. 

Author Response

On behalf of all contributing authors, I sincerely thank the reviewers for their valuable suggestions and comments regarding our manuscript, entitled “Biocompatibility of ABS and PLA Polymers with Dental Pulp Stem Cells: Enhancing their Potential for Biomedical Applications.” We are gratified to note that our work has been recognized as a significant contribution to Biopolymer-Based Biomimetic Scaffolds. In the revised submission, we have meticulously addressed each point raised by the reviewers, as detailed subsequently, and have duly highlighted these modifications within the text of the revised manuscript. We trust that our responses have adequately met the reviewers’ expectations.

In the manuscript, general revisions are highlighted in yellow, while revisions made to the English language are highlighted in cyan blue.

Response to Reviewer 1 comments and suggestions

Point 1.      Abstract: It is well written, but two polymers are used - PLA & ABS. The authors have concluded that both are good, but I advise bringing a parameter to establish a better one in them.

Response 1: We appreciate your suggestion to evaluate which polymer might be more advantageous in certain applications. We aim to demonstrate the benefits of DPSC with both polymers, offering two distinct application options based on the specific requirements. For replacing small tissue fragments, PLA is the ideal choice. Its biodegradable nature allows the body to naturally replace the tissue as the scaffold degrades. In contrast, for extensive bone tissue loss, we recommend using ABS. Given its non-biodegradable properties, ABS remains in the body, providing long-term support where tissue recovery is challenging. We changed the sentences in lines 27, 30-31, and 79-113, which makes this idea clearer.

Point 2.      Introduction: A brief description of the other polymers reported is required to interact with DPSCs. 

Response 2: Addressed. We included in lines 70-72.

Point 3.     Line no. 139/ Is it exactly 30,000?

Response 3: Yes, there are exactly 30,000 cells counted in a Neubauer chamber before plating, line 173. 

Point 4.      2.2: Methods of culturing cells are missing.

Response 4: Addressed. We completed the description of the methodology (lines 162-164 and 167-168).

Point 5.      Figure 1B/ Lane-2: Better to write DPSC+ABS.

Response 5: Addressed. We have changed the figure.

Point 6.      Here, authors can compare both the polymers from the fl. intensity.

Response 6: The quantification of the light signal emitted by bioluminescent cells was analyzed on the same signal intensity scale, equating the analysis of cells when plated on the plate or ABS and PLA polymers with the days evaluated. With this, images that allow visual comparison (Figure 1B) and obtaining values for plotting the data on a graph (Figure 1C) were generated. We make this information clearer on lines 408-410.

Point 7.      3.3: Authors can explain Figure 1B.

Response 7: Figure 1B provides a visual comparison illustrating the dispersion of cells across different surfaces throughout our monitoring period. By comparing the data from day 2 to day 7, we observed an increase in signal distribution (indicating wider cell dispersion) and the intensity of the light signal (suggesting cell proliferation). It's important to note that the light signal is generated by the degradation reaction of the D-Luciferin substrate, a process that only occurs in living cells. Therefore, this data also serves as evidence of cell viability. Additional details on the methodology used are described in lines 201-202.

Point 8.      Discussion: 1st two paragraphs are very general, and that can be shifted to the Introduction.

Response 8. Addressed. The first two paragraphs of the Discussion section have been relocated to the Introduction (now in lines 79-115).

Point 9.  Conclusion: Both polymers were given equal importance. A comparative statement will provide better insights.

Response 9: Addressed. We have already adapted the text (lines 723-727).

Response to Reviewer 2 comments and suggestions

Point 1. Introduce the latest applications of 3D-printed polymers in the biomedical field.

Response 1: Addressed. We have included the latest applications of 3D-printed polymers in the biomedical field in lines 117-124.

Point 2. Explain why ABS and PLA were chosen among many 3D printing materials to study biomedical applications involving DPSCs.

Response 2: Addressed. There has been a growing development in the application of polymers in bioengineering, with the possibility of transcribing medical images (tomography, magnetic resonance imaging) into files for 3D printing, allowing the prototyping of prosthetics customized to the patient's needs. Thinking about polymers that can be applied in 3D printing and that do not have a high cost since the ultimate objective is translational medicine, allowing the creation of anatomical models and prosthetics accessible to the population, we arrived at PLA and ABS. The PLA polymer was already used in the clinic. Still, there was a need to understand whether 3D printing would also present excellent compatibility for producing biodegradable scaffolds. ABS was selected due to its mechanical properties favorable to bone replacement and the fact that it is not absorbable, which would allow another type of application. In addition, there was little description of its application in the literature, making our investigation important. The choice of DPSC for this biological compatibility analysis was due to its easy obtainment, high proliferative rate, and because it is a cell type already used in biocompatibility analyses. The results presented here encourage us to continue a second stage of our research, as we now have another hypothesis to be analyzed whether covering printed models with DPSC can reduce graft rejection, assist in the integration process, and repair the surrounding tissue due to the immunomodulatory and angiogenic action, in addition to the anti-inflammatory, antiapoptotic effects and the release of growth factors. We added the explanation also in lines 117-124.

Point 3. Explain whether the mechanical and thermal properties of ABS and PLA change before and after they are biocompatible with DPSCs, e.g., perform some compression tests on 3D printed scaffolds.

Response 3: This study did not encompass the analysis of mechanical and thermal properties. Given the promising results, future investigations will consider evaluating DPSCs with scaffolds of varied geometries and porosities. In this subsequent research phase, the mechanical and thermal properties assessment is anticipated to yield more applicable and informative insights.

Point 4. Comments on the Quality of English Language. The article needs minor revision for language and grammar.

Response 4: Addressed. We have completed the minor revisions for language and grammar in the article as requested (Highlighted in cyan blue).

Reviewer 2 Report

Comments and Suggestions for Authors

The submitted manuscript investigates the biocompatibility of 3D printed polymers ABS and PLA with dental pulp stem cells. The article is concise and clear. The article has the following comments:

1. Introduce the latest applications of 3D printed polymers in the biomedical field.

2. Explain why ABS and PLA were chosen among many 3D printing materials to study biomedical applications involving DPSCs.

3. Explain whether the mechanical and thermal properties of ABS and PLA change before and after they are biocompatible with DPSCs, e.g., perform some compression tests on 3D printed scaffolds.

Comments on the Quality of English Language

The article needs minor revision for language and grammar.

Author Response

(The authors gave the same response as above.)
